# LLM-Powered Predictive Decision-Making for Sustainable Data Center Operations

## Abstract

The growing demand for AI-driven workloads, particularly from Large Language Models (LLMs), has raised concerns about the significant energy and resource consumption in data centers. This work introduces a novel LLM-based predictive scheduling system designed to enhance operational efficiency while reducing the environmental impact of data centers. Our system utilizes an LLM to predict key metrics such as execution time and energy consumption from source code, and it has the potential to extend to other sustainability-focused metrics like water usage for cooling and carbon emissions, provided the data center can track such data. The predictive model is followed by a real-time scheduling algorithm that allocates GPU resources, aiming to improve sustainability by optimizing both energy consumption and queuing delays. With fast inference times, the ability to generalize across diverse task types, and minimal data requirements for training, our approach offers a practical solution for data center scheduling. This framework demonstrates strong potential for advancing sustainability objectives in AI-driven infrastructure. Through our collaboration with a data center, we achieved a 32% reduction in energy consumption and a 30% decrease in waiting time.

## 1 Introduction

The rapid advancement of Machine Learning (ML), especially Large Language Models (LLMs), has brought about groundbreaking capabilities, yet it has also raised significant social and environmental concerns (Pichai, 2024; Amazon, 2024; Nakagawa & Smith, 2023). One of the most pressing issues is the substantial energy and water resources consumed during ML model training and serving, which has sparked widespread concerns (Blunt & Hiller, 2024; Criddle & Bryan, 2024; Solon, 2021). The computationally intensive ML jobs can demand hundreds of GPU-hours and consume substantial amounts of energy for power and water for cooling. As a result, the growing demand for ML-based applications has significantly amplified the environmental impact of data centers, highlighting the need for modern infrastructure that is both more efficient and sustainable specifically for AI-driven workloads (Bianchini et al., 2024; Li et al., 2024; Kaack et al., 2022). This paper aims to enhance data center operation efficiency and reduce the environmental footprint of modern data centers by devising a LLM-based predictive scheduling system for allocating data center resources (e.g., GPUs) across a stream of ML jobs efficiently. We harness the contextual understanding and predictive abilities of LLMs and combine the predictive power to sequential decision-making that optimize data-center operations.

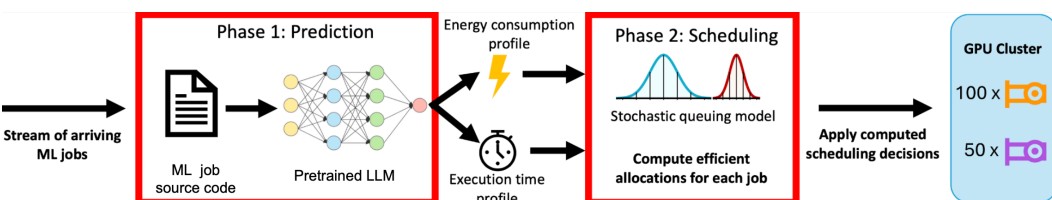

Figure 1: Our Proposed Pipeline

As illustrated in Figure 1, our proposed method first employs an LLM that takes the ML job's source code as input and outputs estimates for the required resources, including execution time and

GPU energy consumption. We believe that the approach has great potential in predicting with other metrics including water consumption and carbon emission, if these data is available from the data center. By leveraging these estimates, the data center can make more informed and efficient real-time decisions for the GPU resource allocation, leading to improvements such as reduced task queue waiting times and lower overall energy consumption.

This work aim to address a key limitation in the current operational pipelines of small to large-scale data centers. We focus on the prevalent practice where users submit tasks and request resources, and the data center allocates resources based on heuristic algorithms that rely on user-provided estimates, such as predicted task duration and resource needs (Tirmazi et al., 2020). The optimal resource allocation, in hindsight, depends on the actual time and energy consumed by the tasks. However, accurately predicting these factors is nearly impossible due to the inherent complexity of modern computing systems. Consequently, neither the user nor the data center has precise knowledge of the optimal resources required for a given task, leading to inefficiencies and resource wastage. Our approach introduces a unified, effective predictive model, followed by a real-time, data-driven scheduling system that enables more efficient and sustainable resource allocation decisions.

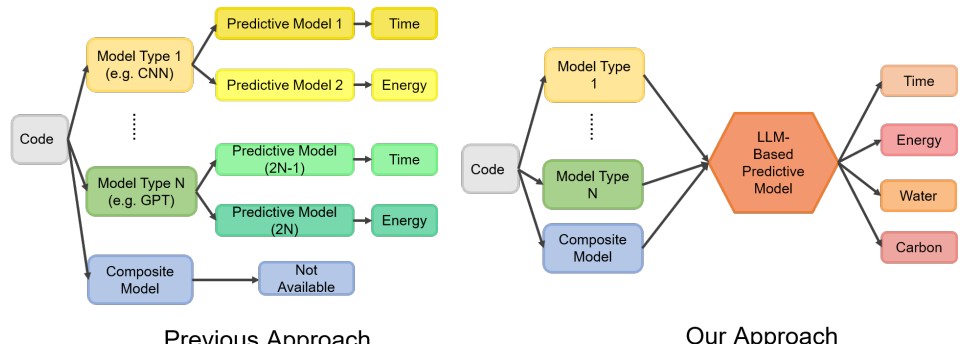

Figure 2: Illustration of our unified predictive approach. Notably, previous methods were unable to handle tasks that had not been seen before, or composite tasks (e.g., training a CNN followed by an LSTM). However, the generalization capacity of LLMs allows our model to effectively manage such cases, making it adaptable to a wider variety of task types and combinations.

## 1.1 OUR CONTRIBUTION

We present a prototype pipeline that provides both methodological and practical contributions to various aspects of data center operations, specifically tailored for AI training and inference applications. Our approach provides methodological insights while opening up new possibilities for applications, especially within today's sustainability and AI-driven context. We defer more discussions on the related works to Appendix A.1.

**Methodological Contributions:** Our work advances the fields of data center predictive modeling and sequential decision-making.

- **LLM-based Versatile Predictive Model:** While many predictive methods exist for data center operations, to our best knowledge, we are the first to offer a comprehensive, end-to-end solution that takes source code as input and outputs estimations of interest. Our model has the potential to be compatible with **any** user-submitted task and can predict **any** measurable metric the data center requires. The strength of our approach lies in the fact that LLM-generated representations are more informative and generalizable than handcrafted features, leading to improved predictive performance and enhanced operational efficiency.

- **Fully Automated Predictive Scheduling System:** This approach enables the possibility of a fully automated predictive scheduling system. By adopting our unified framework, the system is capable of handling any user-submitted tasks and provide estimations of interests directly. This was previously unattainable (see Figure 3), as traditional prediction methods were highly task-specific (e.g., CNN-only or LLM-only), and could only cover a limited

number of task types. Moreover, each type of task required separate prediction models and handcrafted features, significantly limiting the flexibility and scalability of these methods.

- **Sequential Decision-Making Algorithms:** Our predictive framework is complemented by a sequential decision-making algorithm for optimizing resource allocation. This problem arises from the area of reusable resource allocation and queuing control, where no analytical solution exists. We propose a data-driven decision-making algorithm, and implement it in collaboration with a small-size data center. Our algorithm significantly outperforms the baseline scheduling rule, reducing energy consumption by 32% and queuing delays by 30%. This approach leads to more effective data-center management, especially given the growing emphasis on sustainability.

**Application-Level Contributions:** The unique features of our methodology—an end-to-end predictive model and its compatibility with diverse task types and estimation targets—offer substantial potential for developing next-generation AI-driven infrastructures in data centers.

- **Practical Deployment:** Our pipeline executes within one second, enabling a cost-effective, fully automated predictive scheduling system. Additionally, training and deploying the model within data centers is straightforward and can be effective even with a limited amount of data, making it particularly practical for real-world data center operations.

- **Multi-Purpose Sustainable Data Center Operations:** Our method is versatile enough to predict a wide range of metrics, including time and energy, and we believe it can also predict carbon emissions and water consumption. Unlike previous target-specific approaches, which were limited to particular models, our unified framework offers an all-in-one solution that addresses a broad array of predictive needs, providing key insights for building more sustainable and environmentally friendly data center infrastructures.

## 2 PROBLEM FORMATION

**The Prediction Problem.** Consider a data center equipped with various types of GPUs, where machine learning tasks arrive sequentially. The data center must decide which type of GPU to allocate to each arriving task (noting that the number of GPUs is often specified in the user-submitted code or command input). Let $\boldsymbol{x}$ represent the source code submitted by the user, and $z$ represent the data center's decision, which corresponds to the type of GPU to allocate for executing the task. A decision $z$ results in outcomes including the time required to complete the task, denoted by $t$, and the energy consumption using configuration $z$ to complete task $\boldsymbol{x}$, denoted by $e$. The relationship between the outcomes and the decision is modeled by functions $f$ and $g$, such that

$$t := f(\boldsymbol{x}, z), \quad e := g(\boldsymbol{x}, z).$$

Here we note that this function $f$ and $g$ is data-center-specific, it depends on the infrastrcture of the data center, for example, bandwith, CPUs, memory, storage, communication, system setup, software stack, etc. In general, characterizing the functions $f(\boldsymbol{x}, z)$ and $g(\boldsymbol{x}, z)$ is highly challenging due to the complexity of modern computing systems.

**Prediction using Large Language Models.** To tackle the prediction problem, we employ Large Language Models (LLMs) (Ouyang et al., 2022; Radford et al., 2019; Achiam et al., 2023). For simplicity, we refer readers to Vaswani et al. (2017) for a detailed explanation of decoder-based Transformers and LLM architectures. In this paper, we use $f_\theta$ and $g_\theta$ as shorthand notations to represent LLMs, where $\theta$ encapsulates all the model parameters. $f_\theta$ and $g_\theta$ take two inputs: the task's source code $\boldsymbol{x}$ and the data center's decision $z$. The rationale is that the LLM can effectively analyze and interpret the task's source code, transforming it into meaningful representations; these representations, when combined with the GPU configuration $z$, allow the model to produce estimates such that:

$$f_\theta(\boldsymbol{x}, z) \approx f(\boldsymbol{x}, z), \quad g_\theta(\boldsymbol{x}, z) \approx g(\boldsymbol{x}, z).$$

To justify our LLM-based approach, we highlight two key features of LLMs that make them particularly appealing in our context:

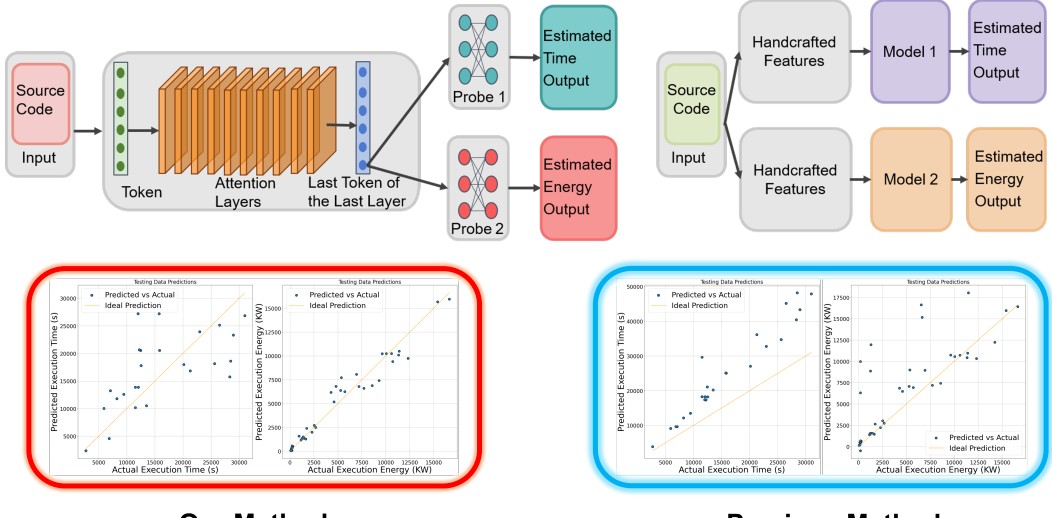

Figure 3: Illustration of our model architecture. Our streamlined design is easy and fast to implement, highly flexible, and generalizable across a variety of tasks. Notice that to estimate other quantities of interest, we simply need to add additional probes. In contrast, previous methods required different handcrafted features and separate prediction models for each task, making them far less flexible and scalable. The lower graph illustrates the performance difference, where points closer to the straight line indicate more accurate predictions. The performance is evaluated under test sets with additional adversarially generated data, and our method outperforms the previous approach (from Cai et al. (2017) and Justus et al. (2018)) in both accuracy and robustness. Notably, the previous method shows consistent systematic bias, likely due to its reliance on "physical" features like the number of forward/backward passes and layer depth, which lack generalizability.

- **Contextual comprehension and feature extraction.** LLMs are renowned for their superior ability to comprehend the context in the source code extract features, named representations. With a large corpus of pre-training data, LLMs and other natural language models (NLP) can discern essential parts of the code with relevant information, transform them into representations containing meaningful information (Tenney et al.; Pilehvar & Camacho-Collados, 2020), and leverage these vectors to predict time, energy consumption, and other metrics.

- **Generalization** Another remarkable capability of LLMs is their ability to generalize (Wei et al., 2022). Even if the pre-training dataset does not cover all possible examples within the function domains, LLMs can still recognize patterns in the dataset and extend their learned knowledge to achieve reasonably accurate approximations.

## 2.1 MODEL ARCHITECTURE - PREDICTION

In this section, we formally describe the architecture of our prediction model. Our approach utilizes a pre-trained LLM for extracting source code representations and applies a probe (Alain, 2016; Radford et al., 2017; Vulić et al., 2020; Zhang et al., 2022) to train a supervised model that predicts the quantity of interest. Figure 3 illustrates the architecture and the different performance compared to previous methods.

- **Extracting Representations:** We adopt a pre-trained LLM to extract task representations, modeled as a function $l_{\theta_1}(\cdot) : \mathcal{V} \to \mathbb{R}^d$, where $\mathcal{V}$ denotes the space of source code and $\mathbb{R}^d$ is the $d$-dimensional embedding space for mapped representations. Here, $\theta_1$ encapsulates the LLM's parameters. Note that $l_{\theta_1}(\cdot)$ refers to the LLM without its final linear and softmax layers, and for the last layer, it outputs the last token's representation, which is a $d$-dimensional vector.

- **Probing:** Probing involves taking the generated representations and predicting the target value using a linear regression or shallow neural network. We denote this probe as $h_{\theta_2}(\cdot, \cdot)$ :

$\mathbb{R}^d \times \mathcal{Z} \to \mathbb{R}$, where $\theta_2$ represents the parameters of the probe and $\mathcal{Z}$ denotes the decision space (e.g., the types of GPUs).

Thus, our predictive model for the execution time can be represented as $f_\theta(\boldsymbol{x}, z) = h_{\theta_2}(l_{\theta_1}(\boldsymbol{x}), z)$. For the energy estimation function $g_\theta(\boldsymbol{x}, z)$, we use the same representation but train a separate probe, denoted $q_{\theta_2}$, such that $g_\theta(\boldsymbol{x}, z) = q_{\theta_2}(l_{\theta_1}(\boldsymbol{x}), z)$. Note that $\theta$, $\theta_1$, and $\theta_2$ refer to classes of parameters, indicating that the parameter dimensions are the same across functions, though these functions do not necessarily share the same parameters.

**Advantage over Previous Methods.** We identify the key factor that could possibly explain the advantage of our method. As mentioned earlier, the challenging part of the data-center operations arise from the complexity of modern computer systems. We first give a notation that characterize the previous approaches.

- For estimating execution time $t = f(\boldsymbol{x}, z)$ and energy consumption $e = g(\boldsymbol{x}, z)$, previous methods rely on handcrafted features, denoted by $\boldsymbol{u}_t = l_t(\boldsymbol{x})$ for time and $\boldsymbol{u}_e = l_e(\boldsymbol{x})$ for energy, where $l_t(\cdot)$ and $l_e(\cdot)$ represent the feature extraction functions. Importantly, the design of $l_t(\cdot)$ and $l_e(\cdot)$ varies significantly depending on the target variable and prediction context, making them different for each prediction task.

- Separate models are then trained for each target, such as $h_t(\cdot, \cdot)$ for time and $h_e(\cdot, \cdot)$ for energy, resulting in the following approximations: $t \approx h_t(l_t(\boldsymbol{x}), z)$ and $e \approx h_e(l_e(\boldsymbol{x}), z)$. To account for the dependence on $z$, which represents the GPU choices, several estimation techniques attempt to model the architecture differences between GPUs. While this approach makes sense physically, it suffers from a lack of flexibility and scalability. Modeling the dependence on $z$ is typically restricted to specific types of source code $\boldsymbol{x}$ and GPU choices $z$, rendering the models unable to generalize to unseen tasks or GPU configurations.

Our approach is better in terms of the architecture for a various reasons

- **Better Representation.** It has been demonstrated in various domains that representations extracted by pre-trained models are significantly more effective than handcrafted features. In this context, our prediction task can also be viewed as a form of feature extraction akin to Natural Language Processing (NLP), where pre-trained models capture richer and more comprehensive information. These representations are more generalizable and universal, making them applicable to a wide variety of prediction tasks related to the characteristics of the source code.

- **Data-Driven Predictive Modeling.** Many traditional predictive models rely on the intrinsic characteristics of the underlying machine learning model, such as CNNs, where predictions are made by explicitly calculating factors like the number of forward/backward passes and the layer depth. While these "physical" models can achieve high accuracy when the test environment perfectly matches the assumptions, they often fall short in data-center operations. For instance, even tasks that train the same CNN architecture may exhibit variations in execution time due to differences in the way the code is written. Additionally, many modern tasks involve training and inference across multiple models, alongside various other function implementations. In such settings, physically based predictive models lack the flexibility and adaptability needed for accurate predictions. In contrast, our data-driven approach is flexible enough to learn the inherent relationships between the source code, the characteristics of computer systems and GPUs, and a variety of targeted metrics.

- **Less Data Hungry.** Traditional methods often require large amounts of source code examples to train predictive models effectively. In contrast, by leveraging a pretrained LLM, which has already been trained on terabytes of data, we capitalize on the model's superior understanding of contextual knowledge. This allows us to obtain a highly effective feature extractor with far less training data, improving both data efficiency and performance. Indeed, our predictive model is trained using only a little more than 500 source code examples, each paired with corresponding results for energy consumption and execution time across different GPUs in the server.

- **Generalization for GPU Dependence.** Another advantage of our approach is its ability to maximize the generalization power of the LLM's representation across different down-

stream tasks. By utilizing the same representation for every prediction task and maintaining a consistent structure for the downstream probe, our method facilitates the discovery of patterns across different GPU configurations. This consistency preserves the model's generalization capacity, ensuring its adaptability to various GPU architectures. Such a factor is missing in previous methods, which followed a more restricted modeling approach, limiting their flexibility and ability to generalize across diverse hardware setups.

**Training Process.** To train our predictive model, for simplicity we assume the parameters of the pretrained LLM $l_{\theta_1}(\cdot)$ remain fixed, and we will discuss the practical training procedure in a followup remark. We first collect a dataset $\mathcal{D}$ consisting of the representations of the source code $l_{\theta_1}(\boldsymbol{x})$ and the corresponding execution results, including time $t$ and energy $e$, across different GPU configurations $z$. Each task $\boldsymbol{x}$ is executed on multiple GPUs to measure both time and energy consumption. Let $|\mathcal{Z}|$ denote the cardinality of the decision space, i.e., the number of available GPUs. The training set, with size $n$, can be written as:

$$\mathcal{D} = \left\{ \left( l_{\theta_1}(\boldsymbol{x}_i), \{t_{ij}, e_{ij}\}_{j=1}^{|\mathcal{Z}|} \right) \right\}_{i=1}^{n},$$

where $t_{ij} = f(\boldsymbol{x}_i, j)$ and $e_{ij} = g(\boldsymbol{x}_i, j)$ are the time and energy required for task $\boldsymbol{x}_i$ when executed on GPU $z = j$. Using this dataset, we train two separate probes: $h_{\theta_2}(\cdot)$ for time prediction , and $q_{\theta_2}(\cdot)$ for energy prediction. The goal is to minimize the error between the actual values $t_{ij}, e_{ij}$ and the predictions $h_{\theta_2}(l_{\theta_1}(\boldsymbol{x}_i), j)$ and $q_{\theta_2}(l_{\theta_1}(\boldsymbol{x}_i), j)$ respectively. Since for simplicity we assume $\theta_1$ is fixed, the loss for the time is $\mathcal{L}_t(\theta_2) = \frac{1}{n|\mathcal{Z}|} \sum_{i=1}^{n} \sum_{j=1}^{|\mathcal{Z}|} (t_{ij} - h_{\theta_2}(l_{\theta_1}(\boldsymbol{x}_i), j))^2$, and the loss for energy prediction is: $\mathcal{L}_e(\theta_2) = \frac{1}{n|\mathcal{Z}|} \sum_{i=1}^{n} \sum_{j=1}^{|\mathcal{Z}|} (e_{ij} - q_{\theta_2}(l_{\theta_1}(\boldsymbol{x}_i), j))^2$.

We remark that the training process described above represents the simplest version for illustrative purposes. In practice, the training process can be more complex and we will address them here.

- One limitation of the approach presented is that the data center would need to run all submitted code to obtain execution time and energy consumption estimates. However, data centers can leverage their existing user submitted codebase and operational data. By recording execution time, energy consumption, and even metrics like carbon emissions and water usage (if measurable), data centers can accumulate vast amounts of data to train the probes effectively. This method allows data centers to continuously refine their predictive models without the need for additional computational overhead.

- Another limitation of the current approach is that we fix the parameters $\theta_1$ of the pretrained LLM and only train the probe parameters $\theta_2$. While this approach simplifies the training process, it may not fully utilize the potential of the model. In practice, we can enhance performance by fine-tuning the LLM or incorporating more advanced techniques like Reinforcement Learning from Human Feedback (RLHF) (Ouyang et al., 2022) or incorporating a reward model (Rafailov et al., 2024). Based on the potential benefits of fine-tuning, RLHF, and reward models, these approaches offer even greater possibilities for improving predictive performance. These enhancements will be considered as part of future work.

## 2.2 PROBLEM FORMATION: DECISION-MAKING

**Constraints.** Consider a data center equipped with $|\mathcal{Z}| = M$ types of GPUs, where tasks $i$ with source code $\boldsymbol{x}_i$ arrive sequentially in time, and the data center must decide which GPU type $z_i \in \mathcal{Z}$ to allocate (notice that the task's source code $\boldsymbol{x}_i$ also specify its preference on GPUs, and we can model this by constraining the action set $\mathcal{Z}$). The GPU resources are limited, and is represented by $\boldsymbol{c} = [c_1, \cdots, c_M]^{\top}$, where $c_j$ is the total number of GPUs of type $j$. If no GPUs are available, the task is placed in a waiting queue. The total available GPU resources at any time $s$ are represented by $\boldsymbol{c}(s) = [c_1(s), \cdots, c_M(s)]^{\top}$, where $c_j(s)$ is the available resource at time $s$ for GPU type $j$.

**Dynamics.** Since all the tasks arrive sequentially, and the task arrival follows a stochastic process, let us denote by $N(T)$ the total number of tasks arrived during time $[0, T]$, and for each task $i$, let $s_i$ represents the arrival time. If there are sufficient non-occupied GPUs available, and we assign $z_i = j$ to task $i$, then $a_{ij}$ number of correponding inventory will be temporary occupied, hence taken out from $c_j(s_i)$. Once $a_{ij}$ of GPU $j$ are assigned to $\boldsymbol{x}_j$, the task will be running for $t_i = f(\boldsymbol{x}_i, z_i)$ amount of time. If we denote by $\mathcal{A}(s)$ the **active set** that contains all tasks that are still running at time $s$, then $i \in \mathcal{A}(s)$ for $s \in (s_i, s_i + t_i)$.

**Waiting Queue.** Let $Q(s)$ denote the set of waiting tasks at time $s$. If a task $\boldsymbol{x}_i$ cannot be immediately processed due to insufficient GPU resources, it is placed in the set $Q(s)$. If task $\boldsymbol{x}_i$ has been put in the queue, we denote by $w_i$ the time it spent in the queue, which depends on factors including other tasks in the waiting queue, the current tasks that are active with their corresponding GPU allocations, and the decision-making model.

**Decision-Making Model.** We aim to find a decision-making policy $\pi$ that takes 4 inputs: the current task feature $\boldsymbol{x}$, the current time $s$, the current inventory level $\boldsymbol{c}(s)$, and the historical information up to the current time $\mathcal{H}_s$. The policy can be deterministic or stochastic, and the range of the policy $\pi$ is $\mathcal{Z} \cup \emptyset$. If $\pi(\boldsymbol{x}, s, \boldsymbol{c}(s), \mathcal{H}) \in \mathcal{Z}$, we assign the task GPU of type $\pi(\boldsymbol{x}, s, \boldsymbol{c}(s), \mathcal{H})$, and if $\pi(\boldsymbol{x}, s, \boldsymbol{c}(s), \mathcal{H}) = \emptyset$, we put this task in the waiting queue. We denote by $\Pi$ the space of policies which satisfies the condition described above. Our goal is to solve the following objective function, which is a combination of the task completion time, the waiting time, and energy cost. (Notice that these times are directly affected by the policy $\pi$, and we denote by $S = \max_{i \in N(T)}\{t_i + w_i\}$ the time upon which all the tasks are finished.)

$$\min_{\pi \in \Pi} \sum_{i=1}^{N(T)} \left( \alpha t_i + \beta w_i + \gamma e_i \right),$$

$$\text{s.t.} \sum_{i=1}^{N(T)} a_{ij} \mathbb{I}(i \in \mathcal{A}(s) \text{ and } z_i = j) \leq c_j, \quad \text{for all } s \in [0, S] \text{ and } j \in \mathcal{Z},$$

where $\alpha$, $\beta$, and $\gamma$ are the weights assigned to execution time, waiting time, and energy cost. This formulation indeed resembles an online allocation problem with reusable resources, as discussed in Chen et al. (2017); Zhang & Cheung (2022). However, two key factors differentiate our work from existing studies in the literature: (i) Multi-purpose objective: while traditional allocation problems typically focus on optimizing a single reward or objective, our approach involves multiple goals. This multi-purpose objective requires balancing various criteria rather than focusing on a singular reward function, adding a layer of complexity not addressed in standard models of online allocation. (ii) Waiting queue: In our problem, there is a waiting queue of tasks that influences decision-making. The policy $\pi$ must account for tasks already in the queue (as inferred from the history $\mathcal{H}$) in addition to handling new arrivals. This contrasts with standard resource allocation models, which usually make decisions based solely on newly arriving tasks, without needing to consider previously queued tasks. This interaction between the queue and decision-making introduces an additional layer of complexity. In summary, these challenges highlight that existing works can not provide (near) optimal solutions for our problem. This necessitates the development of algorithms to effectively address these complexities.

We note that for each task $\boldsymbol{x}_i$, with the predictive model, the data center can incorporate the prediction results of time and energy, $\hat{t}_i := f_\theta(\boldsymbol{x}_i, z_i)$ and $\hat{e}_i := g_\theta(\boldsymbol{x}_i, z_i)$, into their decision-making process. This is because these information can be made available within 1 second after accessing the source code $\boldsymbol{x}_i$, and does not rely on future information. Notice that this decision-making model is also applicable to other objectives, for example, water consumption and carbon emission, if corresponding data is provided by the data center.

We conducted our experiment in collaboration with a data center, collecting two months of operational data from 07/01/2024 to 09/01/2024. As shown in Figure 4 (a), the task arrival times exhibit high non-stationarity. During this period, we recorded the characteristics of tasks, including execution time and energy consumption across different GPUs. This dataset enabled us to backtest our decision-making algorithms that rely on predictions from the LLM-based model.

We propose two algorithms: Greedy (Algorithm 1), where we follow a first-come-first-served rule and the GPU type is selected greedily based on the smallest estimated objective value $\alpha \hat{t}_{ij} + \beta w_i + \gamma \hat{e}_{ij}$, and value-based (Algorithm 2), which is inspired by the algorithm for the multiple knapsack problem (Kan et al., 1993). We compare the performance of these algorithms against the current baseline allocation policy of the data center, with the results presented in Figure 4. Both algorithms outperform the baseline, with the Value-based algorithm achieving a 32% reduction in energy consumption and a 30% reduction in task waiting time. The Value-based method outperforms the Greedy method because it considers all tasks in the waiting queue, and accordingly assigns them based on their "values" of reducing the waiting time and energy cost, rather than following a simple first-come-first-serve rule as in the Greedy algorithm. These results are consistent

with theoretical insights from Spencer et al. (2014); Wagner et al. (2021). More experimental details can be found in Appendix A.3.

| Model | TWT (s) | TDT | CRT(s) | TEC (kWh) |
|---|---|---|---|---|
| **Simple** | 3,135,824.01 | 29.79 | 16,924,533.71 | 470.69 |
| **Value-based** | 2,186,166.49 | 22.50 | 16,565,627.33 | 322.07 |
| **Improvement (%)** | 30.28% | 24.47% | 2.12% | 31.58% |

Table 1: Performance gain for our algorithm implemented in data centers. Here, TWT stands for total waiting time, TDT stands for total delayed tasks, the tasks that have to wait, CRT stands for cumulative running time and TEC stands for the total energy cost.

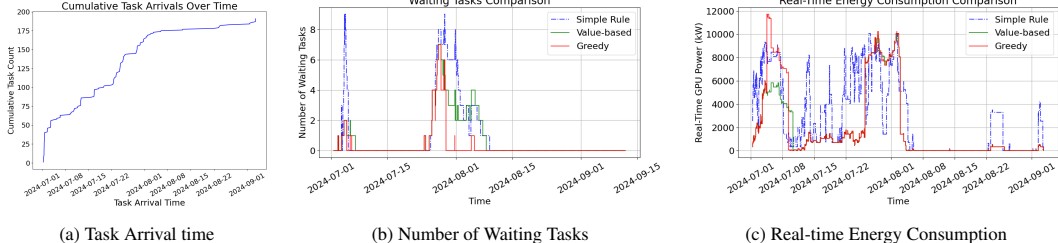

(a) Task Arrival time  (b) Number of Waiting Tasks  (c) Real-time Energy Consumption

Figure 4: (a) Task arrival pattern and (b), (c) performance comparison among the proposed predictive decision-making algorithms, Value-based and Greedy (see Appendix A.3.1), and the benchmark algorithm, Simple Rule. The Simple Rule assigns the available GPU type to the first task in the waiting queue, following a first-come-first-serve policy when sufficient GPUs are available. If multiple GPU types are available, the most powerful one (e.g., A100) is selected.

## 3 OTHER EXTENSION

In this section, we discuss relevant extensions to our pipeline that align with data center practices. Our prototype is designed using a pre-trained LLM with strong capabilities in code comprehension and completion, coupled with a probe trained on 500 source codes. Although we are able to achieve good performance with a relatively small dataset of 500 source codes, this approach still risks poor out-of-sample performance. Due to variations in the coding practices of machine learning engineers—such as differences in variable names, function abstractions, and comments—the contextual information represented in the source code can vary significantly. Even if two codes are functionally equivalent, resulting in identical execution times and energy consumption, the representations generated by the LLM can be quite different.

This issue reflects an inconsistency problem. Formally, there may exist two tasks, $x_1$ and $x_2$, such that the representations differ noticeably ($l_{\theta_1}(x_1) \neq l_{\theta_1}(x_2)$), but the execution times (and energy consumption) are identical: $h_{\theta_2}(l_{\theta_1}(x_1), z) = h_{\theta_2}(l_{\theta_1}(x_2), z)$. We adopt another LLM, referred to as the *Align-LLM*, to address this issue and mitigate potential out-of-distribution (OOD) estimation errors, which could otherwise degrade model performance.

As illustrated in Figure 5, while data centers can gradually accumulate a code base to improve prediction performance for both in-distribution and out-of-distribution tasks, we propose a novel algorithm that partially resolves the OOD issue, showcasing the flexibility and practicality of our pipeline. The architecture and experimental details can be found in appendix A.2

The extended architecture first takes the source code as input and extracts representations using the pre-trained LLM. These representations are compared against representation clusters of the tasks from the local codebase to assess whether the current task's representation deviates significantly from those in the training set. If not, the task is passed to the probe for estimation. Otherwise, the Align-LLM assists by extracting key information from the source code, allowing the data center to identify similar code from its codebase. Next, Align-LLM rewrites the source code based on the style of the similar sample. The rewritten code, although functionally identical to the original, adopts

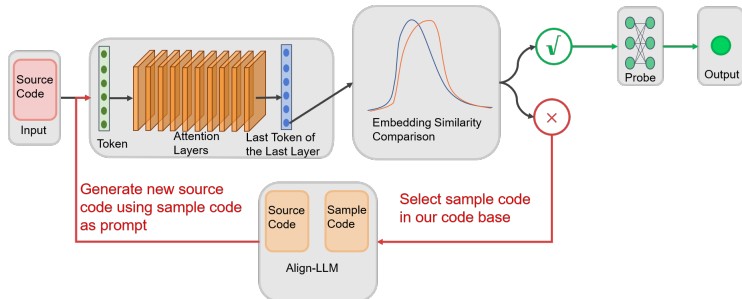

Figure 5: Architecture of the extended pipeline to resolve out-of-distribution tasks

a more familiar written style to the pre-trained LLM, aligning better with the LLM's representations in the training dataset, and enhancing the performance of the downstream probe.

We highlight key insights from our experiments. First, by computing representation distances for the source code in the probe's training dataset, we observe that different task types tend to form distinct clusters (Figure 6 (a)). Second, as shown in Figures 6 (b) and (c), when source code with the same functionality is rewritten by different users or engineers, its embedding diverges from the original cluster, leading to significantly reduced predictive accuracy due to the probe encountering out-of-distribution inputs. However, when we apply Align-LLM to rewrite the code while following the style of the original task, the resulting representation is closer to the original distribution (see indexes 4 vs. 5 and 6 vs. 7 in Figure 6 (b)). Although the rewritten representation is not identical, possibly due to the Align-LLM's style differing slightly from human coding styles, the probe can predict the rewritten code more accurately thanks to the generalizability of our predictive model.

## 4 CONCLUSION

We propose an LLM-based predictive scheduling system for data center operations aimed at enhancing the efficiency and sustainability of AI training. Our system leverages a pre-trained LLM specialized in code comprehension to extract representations of source code and utilizes sequential decision-making algorithms to optimize the scheduling of computational resources. Compared to traditional prediction methods, our architecture design and the use of LLMs offer significantly improved generalizability, flexibility, and practicality. Complemented by the decision-making scheduling algorithms, our approach achieves a 32% reduction in energy consumption and a 30% reduction in waiting times in real data centers. These results demonstrate the strong potential of our method in advancing AI infrastructure.

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

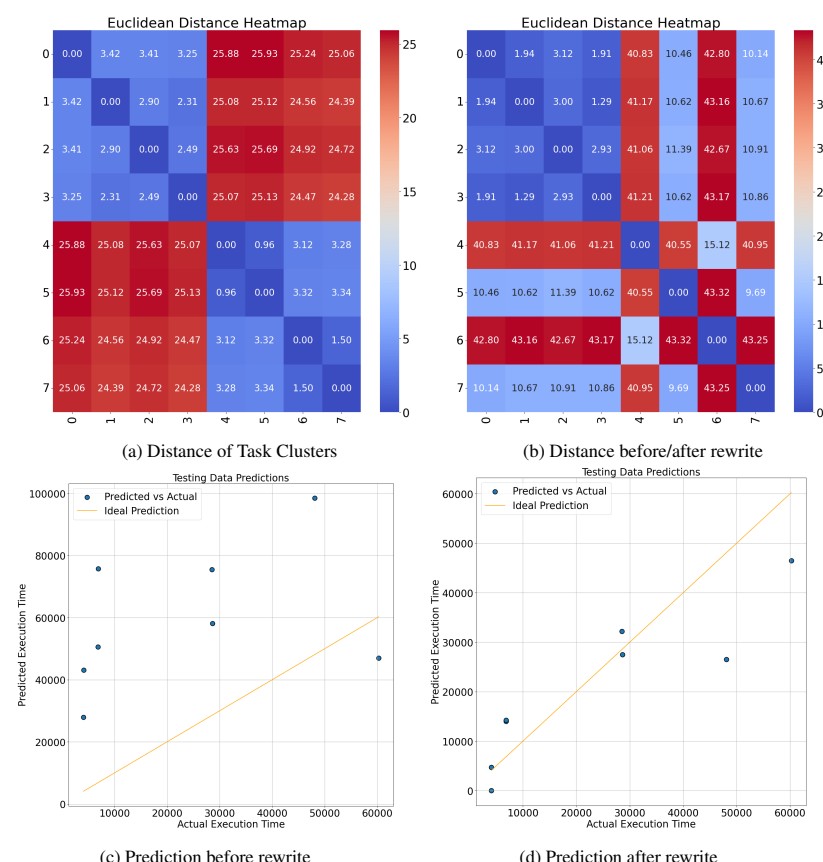

Figure 6: Figure (a) displays the heat map of embedding distances between two classes. Indexes 0–3 represent tasks training the Vit Model, while the remaining indexes correspond to tasks training the GAN Model. In Figure (b), indexes 0 and 1 represent Vit Model training tasks, indexes 2 and 4 represent out-of-distribution Vit Model tasks, and indexes 3 and 6 are LLM-rewritten versions of the task of index 2 and 4. Figures (c) and (d) compare the prediction performance of the execution time for the out-of-distribution tasks before(c) and after(d) the LLM rewrite.

Lu Bai, Weixing Ji, Qinyuan Li, Xilai Yao, Wei Xin, and Wanyi Zhu. Dnnabacus: Toward accurate computational cost prediction for deep neural networks. *arXiv preprint arXiv:2205.12095*, 2022.

Ricardo Bianchini, Christian Belady, and Anand Sivasubramaniam. Datacenter power and energy management: past, present, and future. *IEEE Micro*, 2024.

Katherine Blunt and Jennifer Hiller. Big Tech's latest obsession is finding enough energy. *The Wall Street Journal*, 2024. URL https://www.wsj.com/business/energy-oil/big-techs-latest-obsession-is-finding-enough-energy-f00055b2.

Ermao Cai, Da-Cheng Juan, Dimitrios Stamoulis, and Diana Marculescu. Neuralpower: Predict and deploy energy-efficient convolutional neural networks. In *Asian Conference on Machine Learning*, pp. 622–637. PMLR, 2017.

Qingqing Cao, Yash Kumar Lal, Harsh Trivedi, Aruna Balasubramanian, and Niranjan Balasubramanian. Irene: Interpretable energy prediction for transformers. *arXiv preprint arXiv:2106.01199*, 2021.

Jeffrey S Chase, Darrell C Anderson, Prachi N Thakar, Amin M Vahdat, and Ronald P Doyle. Managing energy and server resources in hosting centers. *ACM SIGOPS operating systems review*, 35 (5):103–116, 2001.

Guanting Chen, Xiaocheng Li, and Yinyu Ye. An improved analysis of lp-based control for revenue management. *Operations Research*, 72(3):1124–1138, 2024.

Yiwei Chen, Retsef Levi, and Cong Shi. Revenue management of reusable resources with advanced reservations. *Production and Operations Management*, 26(5):836–859, 2017.

Eli Cortez, Anand Bonde, Alexandre Muzio, Mark Russinovich, Marcus Fontoura, and Ricardo Bianchini. Resource central: Understanding and predicting workloads for improved resource management in large cloud platforms. In *Proceedings of the 26th Symposium on Operating Systems Principles*, pp. 153–167, 2017.

Cristina Criddle and Kenza Bryan. AI boom sparks concern over Big Tech's water consumption. *The Financial Times*, 2024. URL https://www.ft.com/content/6544119e-a511-4cfa-9243-13b8cf855c13.

Mustafa Daraghmeh, Anjali Agarwal, and Yaser Jararweh. A multilevel learning model for predicting cpu utilization in cloud data centers. In *2023 IEEE Intl Conf on Dependable, Autonomic and Secure Computing, Intl Conf on Pervasive Intelligence and Computing, Intl Conf on Cloud and Big Data Computing, Intl Conf on Cyber Science and Technology Congress (DASC/PiCom/CBDCom/CyberSciTech)*, pp. 1016–1023. IEEE, 2023.

Ding Ding, Xiaocong Fan, Yihuan Zhao, Kaixuan Kang, Qian Yin, and Jing Zeng. Q-learning based dynamic task scheduling for energy-efficient cloud computing. *Future Generation Computer Systems*, 108:361–371, 2020.

Richard Evans and Jim Gao. Deepmind ai reduces google data centre cooling bill by 40 *Google DeepMind blog*, 2016.

Xiaobo Fan, Wolf-Dietrich Weber, and Luiz Andre Barroso. Power provisioning for a warehouse-sized computer. *ACM SIGARCH computer architecture news*, 35(2):13–23, 2007.

Jon Feldman, Monika Henzinger, Nitish Korula, Vahab S Mirrokni, and Cliff Stein. Online stochastic packing applied to display ad allocation. In *European Symposium on Algorithms*, pp. 182–194. Springer, 2010.

Lata J Gadhavi and Madhuri D Bhavsar. Adaptive cloud resource management through workload prediction. *Energy Systems*, 13(3):601–623, 2022.

Sheetal Garg, Rohit Ahuja, Raman Singh, and Ivan Perl. Gmm-lstm: a component driven resource utilization prediction model leveraging lstm and gaussian mixture model. *Cluster Computing*, 26 (6):3547–3563, 2023.

X Yu Geoffrey, Yubo Gao, Pavel Golikov, and Gennady Pekhimenko. Habitat: A {Runtime-Based} computational performance predictor for deep neural network training. In *2021 USENIX Annual Technical Conference (USENIX ATC 21)*, pp. 503–521, 2021.

Eugenio Gianniti, Li Zhang, and Danilo Ardagna. Performance prediction of gpu-based deep learning applications. In *2018 30th International Symposium on Computer Architecture and High Performance Computing (SBAC-PAD)*, pp. 167–170. IEEE, 2018.

Sreenivas Gollapudi and Debmalya Panigrahi. Online algorithms for rent-or-buy with expert advice. In *International Conference on Machine Learning*, pp. 2319–2327. PMLR, 2019.

João Guerreiro, Aleksandar Ilic, Nuno Roma, and Pedro Tomás. Gpu static modeling using ptx and deep structured learning. *IEEE Access*, 7:159150–159161, 2019.

Gül Nihal Güğül, Furkan Gökçül, and Ursula Eicker. Sustainability analysis of zero energy consumption data centers with free cooling, waste heat reuse and renewable energy systems: A feasibility study. *Energy*, 262:125495, 2023.

Muhammad Hafizhuddin Hilman, Maria Alejandra Rodriguez, and Rajkumar Buyya. Task runtime prediction in scientific workflows using an online incremental learning approach. In *2018 IEEE/ACM 11th International Conference on Utility and Cloud Computing (UCC)*, pp. 93–102. IEEE, 2018.

Chen-Yu Hsu, Piotr Indyk, Dina Katabi, and Ali Vakilian. Learning-based frequency estimation algorithms. In *International Conference on Learning Representations*, 2019.

Ling Huang, Jinzhu Jia, Bin Yu, Byung-Gon Chun, Petros Maniatis, and Mayur Naik. Predicting execution time of computer programs using sparse polynomial regression. *Advances in neural information processing systems*, 23, 2010.

Einollah Jafarnejad Ghomi, Amir Masoud Rahmani, and Nooruldeen Nasih Qader. Applying queue theory for modeling of cloud computing: A systematic review. *Concurrency and Computation: Practice and Experience*, 31(17):e5186, 2019.

Daniel Justus, John Brennan, Stephen Bonner, and Andrew Stephen McGough. Predicting the computational cost of deep learning models. In *2018 IEEE international conference on big data (Big Data)*, pp. 3873–3882. IEEE, 2018.

Lynn H Kaack, Priya L Donti, Emma Strubell, George Kamiya, Felix Creutzig, and David Rolnick. Aligning artificial intelligence with climate change mitigation. *Nature Climate Change*, 12(6): 518–527, 2022.

AHG Rinnooy Kan, Leen Stougie, and Carlo Vercellis. A class of generalized greedy algorithms for the multi-knapsack problem. *Discrete applied mathematics*, 42(2-3):279–290, 1993.

Avita Katal, Susheela Dahiya, and Tanupriya Choudhury. Energy efficiency in cloud computing data centers: a survey on software technologies. *Cluster Computing*, 26(3):1845–1875, 2023.

Alexandre Lacoste, Alexandra Luccioni, Victor Schmidt, and Thomas Dandres. Quantifying the carbon emissions of machine learning. *arXiv preprint arXiv:1910.09700*, 2019.

Yanzhe Lei and Stefanus Jasin. Real-time dynamic pricing for revenue management with reusable resources, advance reservation, and deterministic service time requirements. *Operations Research*, 68(3):676–685, 2020.

Xiaocheng Li and Yinyu Ye. Online linear programming: Dual convergence, new algorithms, and regret bounds. *Operations Research*, 70(5):2948–2966, 2021.

Yuzhuo Li, Mariam Mughees, Yize Chen, and Yunwei Ryan Li. The unseen ai disruptions for power grids: Llm-induced transients. *arXiv preprint arXiv:2409.11416*, 2024.

Thodoris Lykouris and Sergei Vassilvitskii. Competitive caching with machine learned advice. *Journal of the ACM (JACM)*, 68(4):1–25, 2021.

Christopher A Metz, Mehran Goli, and Rolf Drechsler. Ml-based power estimation of convolutional neural networks on gpgpus. In *2022 25th International Symposium on Design and Diagnostics of Electronic Circuits and Systems (DDECS)*, pp. 166–171. IEEE, 2022.

Melanie Nakagawa and Brad Smith. On the road to 2030: Our 2022 environmental sustainability report. https://blogs.microsoft.com/on-the-issues/2023/05/10/2022-environmental-sustainability-report/, 2023. Microsoft Blog.

Kenneth O'Neal and Philip Brisk. Predictive modeling for cpu, gpu, and fpga performance and power consumption: A survey. In *2018 IEEE Computer Society Annual Symposium on VLSI (ISVLSI)*, pp. 763–768. IEEE, 2018.

Long Ouyang, Jeffrey Wu, Xu Jiang, Diogo Almeida, Carroll Wainwright, Pamela Mishkin, Chong Zhang, Sandhini Agarwal, Katarina Slama, Alex Ray, et al. Training language models to follow instructions with human feedback. *Advances in Neural Information Processing Systems*, 35: 27730–27744, 2022.

Pratyush Patel, Esha Choukse, Chaojie Zhang, Íñigo Goiri, Brijesh Warrier, Nithish Mahalingam, and Ricardo Bianchini. Characterizing power management opportunities for llms in the cloud. In *Proceedings of the 29th ACM International Conference on Architectural Support for Programming Languages and Operating Systems, Volume 3*, pp. 207–222, 2024.

Thanh-Phuong Pham, Juan J Durillo, and Thomas Fahringer. Predicting workflow task execution time in the cloud using a two-stage machine learning approach. *IEEE Transactions on Cloud Computing*, 8(1):256–268, 2017.

Sundar Pichai. Climate change is humanity's next big moonshot. `https://blog.google/outreachinitiatives/sustainability/dear-earth/`, 2024. Google Blog.

Mohammad Taher Pilehvar and Jose Camacho-Collados. *Embeddings in natural language processing: Theory and advances in vector representations of meaning*. Morgan & Claypool Publishers, 2020.

Eduardo Pinheiro, Ricardo Bianchini, Enrique V Carrera, and Taliver Heath. Load balancing and unbalancing for power and performance in cluster-based systems. 2001.

Manish Purohit, Zoya Svitkina, and Ravi Kumar. Improving online algorithms via ml predictions. *Advances in Neural Information Processing Systems*, 31, 2018.

Alec Radford, Rafal Jozefowicz, and Ilya Sutskever. Learning to generate reviews and discovering sentiment. *arXiv preprint arXiv:1704.01444*, 2017.

Alec Radford, Jeffrey Wu, Rewon Child, David Luan, Dario Amodei, Ilya Sutskever, et al. Language models are unsupervised multitask learners. *OpenAI blog*, 1(8):9, 2019.

Rafael Rafailov, Archit Sharma, Eric Mitchell, Christopher D Manning, Stefano Ermon, and Chelsea Finn. Direct preference optimization: Your language model is secretly a reward model. *Advances in Neural Information Processing Systems*, 36, 2024.

Parthasarathy Ranganathan, Phil Leech, David Irwin, and Jeffrey Chase. Ensemble-level power management for dense blade servers. *ACM SIGARCH computer architecture news*, 34(2):66–77, 2006.

Charlotte Rodriguez, Laura Degioanni, Laetitia Kameni, Richard Vidal, and Giovanni Neglia. Evaluating the energy consumption of machine learning: Systematic literature review and experiments. *arXiv preprint arXiv:2408.15128*, 2024.

Olivia Solon. Drought-stricken communities push back against data centers. *NBC News*, 2021. `https://www.nbcnews.com/tech/internet/drought-stricken-communities-push-back-against-data-centers-n1271344`.

Joel Spencer, Madhu Sudan, and Kuang Xu. Queueing with future information. *ACM SIGMETRICS Performance Evaluation Review*, 41(3):40–42, 2014.

Ian Tenney, Patrick Xia, Berlin Chen, Alex Wang, Adam Poliak, R Thomas McCoy, Najoung Kim, Benjamin Van Durme, Samuel R Bowman, Dipanjan Das, et al. What do you learn from context? probing for sentence structure in contextualized word representations. In *International Conference on Learning Representations*.

Kundjanasith Thonglek, Kohei Ichikawa, Keichi Takahashi, Hajimu Iida, and Chawanat Nakasan. Improving resource utilization in data centers using an lstm-based prediction model. In *2019 IEEE international conference on cluster computing (CLUSTER)*, pp. 1–8. IEEE, 2019.

Muhammad Tirmazi, Adam Barker, Nan Deng, Md E Haque, Zhijing Gene Qin, Steven Hand, Mor Harchol-Balter, and John Wilkes. Borg: the next generation. In *Proceedings of the Fifteenth European Conference on Computer Systems*, pp. 1–14, 2020.

Shreshth Tuli, Shashikant Ilager, Kotagiri Ramamohanarao, and Rajkumar Buyya. Dynamic scheduling for stochastic edge-cloud computing environments using a3c learning and residual recurrent neural networks. *IEEE transactions on mobile computing*, 21(3):940–954, 2020.

Ashish Vaswani, Noam Shazeer, Niki Parmar, Jakob Uszkoreit, Llion Jones, Aidan N Gomez, Łukasz Kaiser, and Illia Polosukhin. Attention is all you need. *Advances in neural information processing systems*, 30, 2017.

Jordi Vilaplana, Francesc Solsona, Ivan Teixidó, Jordi Mateo, Francesc Abella, and Josep Rius. A queuing theory model for cloud computing. *The Journal of Supercomputing*, 69:492–507, 2014.

Ivan Vulić, Edoardo Maria Ponti, Robert Litschko, Goran Glavaš, and Anna Korhonen. Probing pretrained language models for lexical semantics. In *Proceedings of the 2020 Conference on Empirical Methods in Natural Language Processing (EMNLP)*, pp. 7222–7240, 2020.

Benjamin Wagner, André Kohn, and Thomas Neumann. Self-tuning query scheduling for analytical workloads. In *Proceedings of the 2021 International Conference on Management of Data*, pp. 1879–1891, 2021.

Haifeng Wang and Yunpeng Cao. Predicting power consumption of gpus with fuzzy wavelet neural networks. *Parallel Computing*, 44:18–36, 2015.

Jason Wei, Yi Tay, Rishi Bommasani, Colin Raffel, Barret Zoph, Sebastian Borgeaud, Dani Yogatama, Maarten Bosma, Denny Zhou, Donald Metzler, et al. Emergent abilities of large language models. *Transactions on Machine Learning Research*, 2022.

Qiang Wu, Qingyuan Deng, Lakshmi Ganesh, Chang-Hong Hsu, Yun Jin, Sanjeev Kumar, Bin Li, Justin Meza, and Yee Jiun Song. Dynamo: Facebook's data center-wide power management system. *ACM SIGARCH Computer Architecture News*, 44(3):469–480, 2016.

Qiaomin Xie and Yi Lu. Priority algorithm for near-data scheduling: Throughput and heavy-traffic optimality. In *2015 IEEE Conference on Computer Communications (INFOCOM)*, pp. 963–972. IEEE, 2015.

Kuang Xu and Carri W Chan. Using future information to reduce waiting times in the emergency department via diversion. *Manufacturing & Service Operations Management*, 18(3):314–331, 2016.

Lining Zhang, Mengchen Wang, Liben Chen, and Wenxin Zhang. Probing gpt-3's linguistic knowledge on semantic tasks. In *Proceedings of the Fifth BlackboxNLP Workshop on Analyzing and Interpreting Neural Networks for NLP*, pp. 297–304, 2022.

Xilin Zhang and Wang Chi Cheung. Online resource allocation for reusable resources. *arXiv preprint arXiv:2212.02855*, 2022.

Qi Zhao, Hailong Yang, Zhongzhi Luan, and Depei Qian. Poigem: A programming-oriented instruction level gpu energy model for cuda program. In *Algorithms and Architectures for Parallel Processing: 13th International Conference, ICA3PP 2013, Vietri sul Mare, Italy, December 18-20, 2013, Proceedings, Part I 13*, pp. 129–142. Springer, 2013.

Meiling Zhou, Jie Chen, Haiyang Hu, Jiacheng Yu, Zhongjin Li, and Hua Hu. Deeptle: Learning code-level features to predict code performance before it runs. In *2019 26th Asia-Pacific Software Engineering Conference (APSEC)*, pp. 252–259. IEEE, 2019.

# A APPENDIX

## A.1 DISCUSSION ON RELATED WORKS

**Prediction based on device features.** A body of literature focuses on predicting relevant metrics, such as execution time and energy consumption, using features that summarize the characteristics of the hardware. This approach assumes that analyzing hardware parameters and runtime data can uncover patterns that influence these metrics. For example, Pham et al. (2017) combine GPU runtime parameters with static hardware features, applying regression models to predict execution time. Similarly, Daraghmeh et al. (2023) and Garg et al. (2023) utilize various clustering techniques to identify operational patterns in machines based on hardware metrics, followed by sequence modeling for accurate predictions. Conversely, Hilman et al. (2018) take an alternative approach by first predicting hardware behavior during code execution and then employing the KNN clustering method to forecast execution time. In general, this stream of literature is not very related to our approach and we refer the readers to O'Neal & Brisk (2018); Rodriguez et al. (2024); Ali et al. (2023) for literature survey.

**Prediction based on code features.** Regarding code characteristics, early work by Huang et al. (2010) used feature engineering, extracting elements like loop counts and conditional branches, and applying sparse polynomial regression for time prediction. Recent approaches have shifted towards deep learning, where two main strategies dominate. One approach treats models as composed of atomic operations, with works like Wang & Cao (2015); Cai et al. (2017); Geoffrey et al. (2021); Justus et al. (2018) using program slicing and MLPs to predict time and energy by decomposing models into layers. The second approach leverages graph-based techniques, as in Cao et al. (2021) and Bai et al. (2022), which use graphs to represent layer dependencies and employ machine learning to learn these representations. Some methods are similar to ours in extracting code representations for prediction. For example, Guerreiro et al. (2019) transforms PTX instructions into embeddings for LSTM inputs, while Zhou et al. (2019) uses attention-based Bi-LSTMs and graph convolutional networks to automatically extract code semantics and structure. While these methods focus on extracting features, our model generalizes across a wider variety of task types and prediction metrics using a unified, high-level representation approach. See Gianniti et al. (2018); Metz et al. (2022); Zhao et al. (2013) for more related work.

**Data center operations.** Energy management in data centers has been a longstanding area of interest, with foundational work by Pinheiro et al. (2001); Chase et al. (2001); Ranganathan et al. (2006); Fan et al. (2007). As cloud computing and AI technologies emerged, software-based approaches have been developed to improve data center operations (Wu et al., 2016; Evans & Gao, 2016; Cortez et al., 2017; Katal et al., 2023). In today's AI and sustainability-driven era, there has been growing interest in carbon emissions related with AI (Lacoste et al., 2019; Anderson et al., 2023; Güğül et al., 2023; Patel et al., 2024). However, most efforts focus on software or infrastructure-level operations, whereas our approach specifically targets algorithmic-level improvements. While there is a substantial body of literature on optimal scheduling and queuing policies for service systems, much of this work is highly theoretical and relies on numerous assumptions (Vilaplana et al., 2014; Xie & Lu, 2015; Jafarnejad Ghomi et al., 2019). Additionally, many scheduling algorithms that avoid theoretical assumptions, such as reinforcement learning-based approaches (Ding et al., 2020; Tuli et al., 2020), primarily focus on scheduling tasks but lack task-level predictive inference capabilities, which is a core strength of our method. Additionally, we note that our focus is specifically on the operations of data centers for AI-driven workloads, which exhibit distinct characteristics compared to traditional data center operations considered in previous studies.

**Predictive decision-making** Our work is also related to the area of decision-making with future predictions as side information. With such predictions, well-established decision algorithms, originally designed without the benefit of foresight, can be improved (Purohit et al., 2018). Examples include problems such as caching (Lykouris & Vassilvitskii, 2021), rent-or-buy (also known as the ski rental problem) (Gollapudi & Panigrahi, 2019), frequency estimation (Hsu et al., 2019), and queuing control (Spencer et al., 2014; Xu & Chan, 2016). Among these, the most closely related topic to our work is (online) resource allocation (Feldman et al., 2010; Li & Ye, 2021; Chen et al., 2024; Zhang & Cheung, 2022). Specifically, Lei & Jasin (2020); Chen et al. (2017) examine online allocation with reusable resources (like GPUs in data centers), where each arriving request occupies resources for only a limited period before releasing them. Thonglek et al. (2019) and Gadhavi &

Bhavsar (2022) focus on CPU and memory utilization, adopting LSTM and other time series models to optimize resource allocation.

## A.2 EXPERIMENT: PREDICTIVE MODEL DETAILS

### A.2.1 EXPERIMENT SETUP

**Data Generation**  The training (and testing) data for the probes includes embedding features and corresponding label values.

- **Embedding Features**: The input features for the probe model are generated through Starcoder-7B(LLM). Specifically, we input 500 source code files as prompts into Starcoder-7B(LLM) and extract 4608-dimensional embedding vectors from each inference's penultimate layer, i.e., the output of the last transformer block. Following standard practices for sequence classification tasks, we then use the last token's embedding vector to represent the features of the entire code file. The 500 code files are either carefully selected or handcrafted to ensure that each file can run independently to complete a full computational process. These files cover a diverse set of tasks and structures, including ResNet, BERT, GAN, ViT, VGG, and more. The minimum, average, and maximum number of tokens in the code files are 1037, 1582.26, and 2151, respectively. And We ensure that the LLM's context length is sufficient to process the entire code file without truncation.

- **Label Values**: The label values, which the probes aim to predict, are generated by running the 500 code files on two different types of GPUs, NVIDIA A100 and NVIDIA A6000. We record the running time and energy consumption using the official tool `nvidia-smi` by NVIDIA. For each code running, we open an independent process running `nvidia-smi --query-gpu=power.draw` to record the real-time power consumption with a logging interval of 1 second and compute the average power consumption. For each code file, we run experiments for at least twice and make sure the gap of recorded values is less than 10% of the average. We also make sure the gpus are exclusively used by our experiments.

**Probe Architecture**  In our experiments, the probe model is a 3-layer dense neural network, utilizing ReLU as the activation function, with batch normalization applied to each layer. The input dimension of 4608 aligns with the dimensions of the (input) embedding vectors. The embedding dimensions for each layer are 1024, 30, and 1, respectively.

**Probe Training**  We randomly separate the generated data into training data and testing data with a ratio of 9:1. All the data are further normalized by Standardscaler. The probe models are trained using the following configurations: a batch size of 8, 2000 training epochs, a learning rate of 1e-4, weight decay of 0.001, and $L1$ regularization with penalty parameter being 1e-5. We use the Mean Squared Error (MSE) as the loss function and AdamW as the optimizer. Additionally, we apply early stopping when the epoch loss decreases by no more than 0.001 for 30 consecutive epochs.

**Testing and Inference**  To test the time needed for prediction, in the testing phase we let the LLM Starcoder take the source code as input, outputting the presentation, and use the probe to predict the estimated value. The inference time for 48 source codes takes 32 seconds, averaging 0.65 seconds per task. The testing phase is carried out on 1 Nvidia RTX 6000 (Ampere Version) GPU.

**Predictive pipeline for OOD**  For the experiments in section 3, we take gpt-4o-2024-05-13 as the Align-LLM. With the Align-LLM, the rewrite time for the source code of 7 tasks takes 68 seconds, averaging 9.7 seconds per task.

## A.3 EXPERIMENT: DECISION-MAKING MODEL DETAILS

In this section, we first provide detailed predictive decision-making algorithms applied in A.3. Further, due to data privacy restrictions imposed by the collaborating data center, we present additional numerical results across various settings using a simulation system. These results can offer managerial insights for the data center operator and validate our choice of multi-criteria optimization.

### A.3.1 PREDICTIVE DECISION-MAKING ALGORITHMS

In this section, we provide detailed implementations of the two types of predictive decision-making algorithms used for GPU allocation.

---

**Algorithm 1** Predictive Decision-Making: Greedy

---

**Require:** Current time $s$, GPU types $\mathcal{Z}$, waiting queue $Q(s)$, active set $\mathcal{A}(s)$, GPU occupation information $a_{ij}$'s, available GPUs $\boldsymbol{c}(s)$, prediction models $f_\theta$, $g_\theta$, weights $\alpha, \gamma$.

1: Estimate the running time and the energy of each task $i \in \mathcal{A}(s) \bigcup Q(s)$ for each GPU type $j \in \mathcal{Z}$:
$$\hat{t}_{ij} = f_\theta(\boldsymbol{x}_i, j), \quad \hat{e}_{ij} = g_\theta(\boldsymbol{x}_i, j)$$
%% First-come-first-served rule

2: Sort $Q(s)$ in ascending order by their arriving time $s_i$, and assign the first task (which first comes) to $z_1 = \arg\min_{j \in \mathcal{J}} \alpha \hat{t}_{1j} + \gamma \hat{e}_{1j}$ whenever $z_1$ has sufficient GPUs to satisfy task 1, i.e., at the time $\min\{s' \geq s | \boldsymbol{c}_{z_1}(s') \geq a_{1z_1}\}$

---

**Algorithm 2** Predictive Decision-Making: Value Based

---

**Require:** Current time $s$, GPU types $\mathcal{Z}$, waiting queue $Q(s)$, active set $\mathcal{A}(s)$, GPU occupation information $a_{ij}$'s, available GPUs $\boldsymbol{c}(s)$, prediction models $f_\theta$, $g_\theta$, hyperparameter $\kappa$.

1: Estimate the running time and the energy of each task $i \in \mathcal{A}(s) \bigcup Q(s)$ for each GPU type $j \in \mathcal{Z}$:
$$\hat{t}_{ij} = f_\theta(\boldsymbol{x}_i, j), \quad \hat{e}_{ij} = g_\theta(\boldsymbol{x}_i, j)$$

2: Compute the value $v_{ij}$ for each task $i \in Q(s)$ and each GPU type $j \in \mathcal{Z}$:

$$v_{ij} = \frac{1}{a_{ij}\hat{t}_{ij}} - \kappa \hat{e}_{ij}$$

3: Construct the value set $\mathcal{V} = \{(i, j, v_{ij})\}$ and sort it in decending order by the cost value $v_{ij}$.
4: **for** $(i, j, v_{ij}) \in \mathcal{V}$ **do**
5:     **if** $a_{ij} \leq \boldsymbol{c}_j(s)$ **then**
    %% If there are sufficient GPUs, assign $i$ to $j$
6:         Set $z_i = j$, and update $\mathcal{V}$ by removing the tuples $\{(i', j', v_{i'j'}) \in \mathcal{V} | i' = i, j' \in \mathcal{Z}\}$
7:         Update the available GPUs $\boldsymbol{c}_j(s) = \boldsymbol{c}_j(s) - a_{ij}$
8:     **else**
9:         Skip
10:     **end if**
11: **end for**

---

The Greedy algorithm focuses on a first-come-first-served approach to allocate tasks to GPUs. At each time step $s$, the algorithm estimates the running time and energy consumption for each task $i$ on each GPU type $j$ using the prediction models. Once these estimates are made, the waiting queue is sorted in ascending order of task arrival times. The algorithm assigns the first task in the queue to the GPU type selected based on the smallest estimated value of $\alpha \hat{t}_{1j} + \gamma \hat{e}_{1j}$ reflecting a weighted combination of running time and energy consumption. This algorithm's simplicity makes it efficient for quick decision-making, but it may not always optimize resource allocation across the entire queue.

The Value-Based algorithm extends beyond the Greedy approach by considering all tasks in the waiting queue and selecting the GPU allocation based on a more strategic optimization. At each time step the algorithm first estimates the running time and energy consumption for each task across GPU types. However, the next step involves computing a cost value $v_{ij}$ for each task-GPU pair, which is a combination of the inverse of the estimated running time $\hat{t}_{ij}$ adjusted by the GPU occupancy requirement $a_{ij}$ and a penalty term $\kappa \hat{e}_{ij}$ representing the energy consumption. The algorithm then constructs a set of task-GPU pairs, sorting them in descending order based on the value $v_{ij}$. It allocates the GPUs to tasks based on this sorted list, prioritizing higher values, and updates the available GPU resources accordingly. This method is inspired by the algorithms in multiple knap-

sack problems and balances multiple objectives, such as reducing energy consumption and running time with limited available GPUs.

### A.3.2 More Simulation Experiments

**Simulation Environment**  We built the simulation environment using data collected from the co-operating data center. Specifically, we first estimated a heterogeneous Poisson process to model task arrivals. And we build a task simulator which generates the running time $t_{ij}$ and the energy consumption $e_{ij}$ for each arrival task $i$ across GPU type $j$. The values are randomly sampled from truncated normal distributions (truncated above 0), with the mean and variance estimated from the collected data. Further, we assume there are two types of GPUs ($|\mathcal{Z}| = 2$) with the number of each type randomly sampled uniformly from the intervals $[10, 20]$ and $[20, 40]$, respectively.

**Performance under different criteria**  We evaluate the performance of the proposed Algorithm 2 under different settings, with varying emphasis on the optimization criteria. Specifically, we tune the hyperparameter $\kappa$ in Algorithm 2 using independently sampled validation data, based on the following metrics: (i) waiting time only ($\alpha = \gamma = 0$ and $\beta = 1$) (ii) running time only ($\beta = \gamma = 0$ and $\alpha = 1$), and (iii) energy only ($\alpha = \beta = 0$ and $\gamma = 1$). These settings prioritize different objectives, allowing us to compare the results and validate the effectiveness of the proposed Algorithm 2 under various criteria. We compare the performance of Algorithm 2 with a simple rule, which assigns the available GPU type to the first task in the waiting queue, following a first-come-first-serve policy, provided there are sufficient GPUs. When multiple GPU types are available, the most powerful type (e.g., A100) is selected. The reported results are based on 100 testing samples and the validate data contains 50 samples.

Table 2 summarizes the experimental results. First, it showcases that the proposed value-based rule can outperform the benchmark simple rule consistently in all tuning methods. In addition, it demonstrates that the performance of the value-based rule aligns with the specific objective being emphasized during tuning. Specifically, when tuned to minimize waiting time, it achieves the shortest waiting time (and the fewest tasks with positive wait times). Similarly, when tuned to minimize running time, it achieves the smallest running time. Finally, when tuned to minimize energy consumption, it achieves the lowest energy usage during testing. We also provide visualizations of sample path levels in Figure 7.

| **Metric** | Simple rule | Value-based rule | | |
| --- | --- | --- | --- | --- |
| | | Waiting time | Running time | Energy |
| **Total Waiting Time (s)** | 2,466,062.56 | **1,810,123.92** | 2,045,399.25 | 2,112,773.56 |
| -Tasks with Wait Time | 27.38 | **21.16** | 22.13 | 22.80 |
| **Total Running Time (s)** | 15,333,868.47 | 15,215,721.21 | **15,001,371.15** | 15,291,965.57 |
| **Total Energy Cost (kWh)** | 440.98 | 245.42 | 243.97 | **240.07** |

Table 2: Comparison of testing total waiting time, tasks with wait time, cumulative running time, and energy cost for the benchmark algorithm (simple rule) and value-based models tuned under different emphasized objectives.

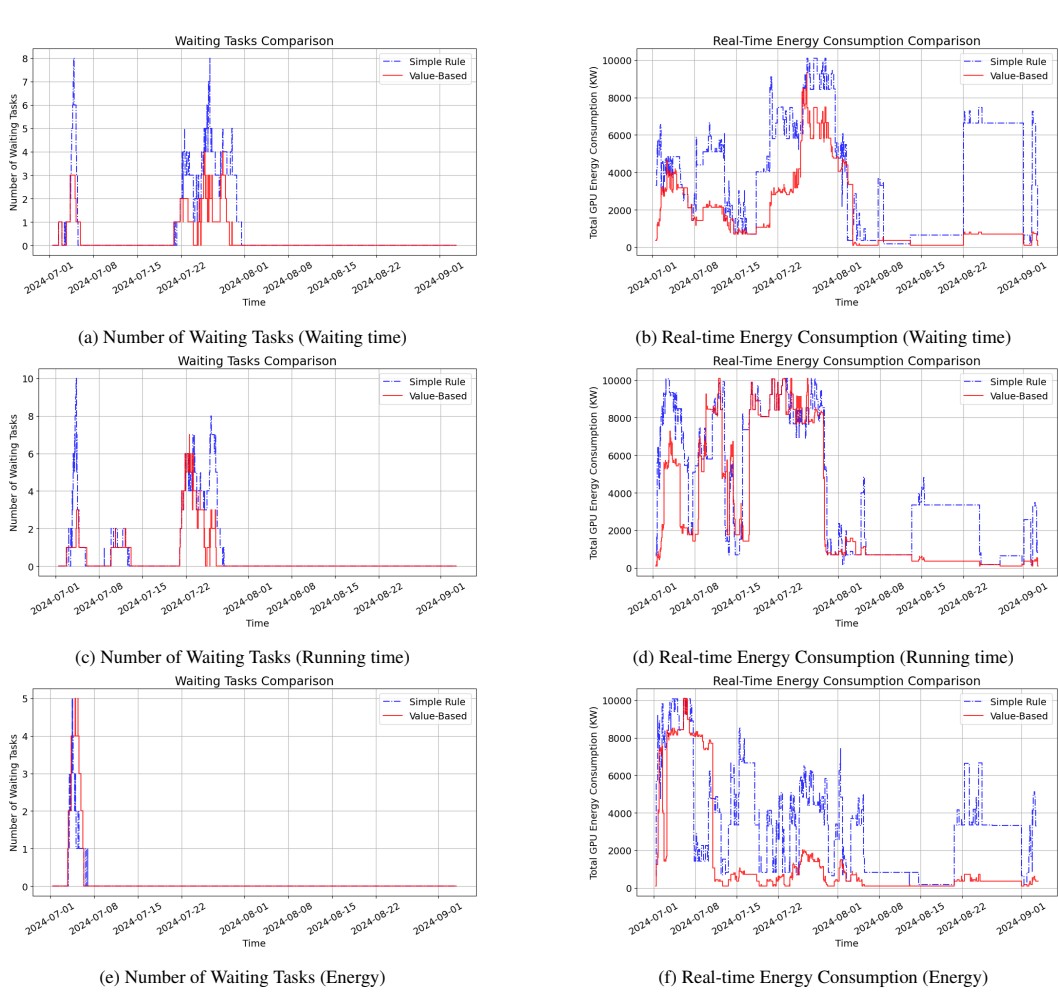

Figure 7: Performance comparison across different metrics for the benchmark algorithm (simple rule) and value-based models tuned under different emphasized objectives.

