# OpenReview forum: "LLM-Powered Predictive Decision-Making for Sustainable Data Center Operations"
_ICLR.cc/2025/Conference — ICLR 2025 Conference Withdrawn Submission_

### Official Review · Reviewer_vdc1 · 2024-10-28

**Soundness:** 2
**Presentation:** 3
**Contribution:** 1
**Rating:** 3
**Confidence:** 5

**Summary:**

The paper introduces a compound-AI based technique to predict the performance, energy consumption, and other key operational metrics for data center operations. The paper motivates the problem well, showing how a pre-trained LLM can possibly be used to predict the performance of workloads on different hardware types. This can be further used in scheduling of workloads on devices. The authors then device a scheduling optimization problem along with two algorithms to show how such a deployment can help datacenter operators. The authors run simulations based on a dataset acquired from a production system from a small datacenter over a period of about 2 months. The dataset has an aggregate task counts of less than 200 tasks. They adapt the pretrained model using 500 source codes. To label the data (and run their experiments), the authors use two GPU models A100 and A6000.

**Strengths:**

1. Thank you for submitting your paper to ICLR. I enjoyed reading the paper as it is well written generally.
2. The paper covers an important topic that many datacenter operators care about, how to better utilize accelerator resources.
3. The paper uses data from a data center and I believe is the first paper to suggest a compound AI system with two LLMs to assign GPU resources.

**Weaknesses:**

I think the paper however has several shortcomings that I will aim to detail next. The paper is neither strong on the systems side nor on the ML side, and this is the main shortcoming in my opinion. I will detail what I mean by that in the next points:
1. To start with, I am not entirely sure if for the scale of the problem you define, the LLM is doing much. For an ICLR submission, I think It would have been better to focus more on the ML side of the problem and not the decisions making. After all, you have only provided an overview of the prediction results in the paper in Table 1. However, non of these results show tradtitional metrics that one would expect on the predictions, e.g., accuracy, recall, F1-Score, MAPE, etc. I would like to see some of these aspects for the method.
2. There is not much novelty in the ML side of the paper, except maybe with the Align-LLM part. However, the authors treat this in passing only, with very little to no evaluations on even how this extra LLM helps. It would help the paper to do an ablation study with Align-LLM. In addition, you effectively have only two classes for your classifier, A100 and A6000. I wonder how your system would expand to a larger system with say 10s of accelerator types?
3. From a systems perspective, I think there are way too many shortcomings. Since ICLR is not a systems conference, I will only include the following few shortcomings. First of all, you have a total of less than 200 tasks over a period of 2 month. That is about three tasks per day. Since you are running this in simulations, you can scale this up by, e.g., creating a set that resembles the original tasks you have. There are also multiple other public traces now available, e.g., from Alibaba with GPU workloads (https://github.com/alibaba/clusterdata/tree/master/cluster-trace-gpu-v2020#pai_task_table) . That being said, you do not need even GPU traces, you can easily simluate larger systems.

- Second, what is your use-case? A task that runs for short periods? How would you know how long this task runs in a real datacenter unless it is a repetitive workload? Third, how would your system scale with 30+ accelerator types and 10s to 100s of tasks arriving per minute, per second, and per hour?

**Questions:**

1. My first question is, what is the use-case that you are trying to solve?
2. What are the accuracy and prediction metrics of your system?
3. What is the scale of the datacenter you collaborate with?

---

### Official Review · Reviewer_Hxm4 · 2024-11-03

**Soundness:** 2
**Presentation:** 3
**Contribution:** 3
**Rating:** 3
**Confidence:** 4

**Summary:**

This paper presents an LLM-powered automatic predictive scheduling system for  AI workloads in data center, with the goal of optimizing both performance(job completion time) and energy consumption. The system consists of two main components: 1) An LLM-based predictive model that takes job's source code as input, and predicts its execution time and energy consumption; 2) A decision-making model that uses these predictions for deciding GPU resource allocation to each job. Through collaboration with a data center, the authors demonstrated 32% reduction in energy consumption and 30% decrease in waiting time. The key innovation is using LLMs to generate code representations that enable generalizable prediction across diverse task types.

**Strengths:**

The paper designs a novel end-to-end solution using LLMs for predictive data center resource allocation. Also, their combination of LLM and probe network reduces the number of training data needed.

Their framework can generalize to diverse job task types including composite and unseen tasks, making it more flexible than traditional methods that required separate models for different task types.

The writing is easy to follow and clearly explains their model architecture.

**Weaknesses:**

The paper lacks information about which pre-trained LLM was used, details about its output representation, and how did they leverage LLM to generate the output representation (eg, prompting method).

The evaluation section seems to be incomplete. More comprehensive evaluation details are necessary to evaluate whether their proposed solution works.

The proposed method could cause potential privacy concerns when sending confidential user-submitted code to LLM for analysis.

No discussion of the computational and cost overhead of running the LLM-based prediction framework.

**Questions:**

Thank you for submitting to ICLR. I really like the idea of using LLMs for predictive data center resource allocation. However, the paper seems incomplete and lacks a robust evaluation for the proposed method. Addressing this issue would strengthen the paper.


Questions:

Could you provide more details about the LLM model used in the experiments, including
- LLM model type, and how was it pre-trained or fine-tuned for this task
- the dimension of the LLM output representations
- how to leveage LLM to generate the output representation (eg, prompting method)

What is the computational and cost overhead of running the LLM-based prediction framework? How does this compare to the benefits gained in terms of improved resource allocation and reduced energy consumption?

Can you provide more details about the evaluation, including
- detailed experimental setup
- data center scale
- the number and distribution of different task types
- baseline scheduling algorithms to compare with
- evaluation metrics
- ablation study

How does the framework handle prediction errors? Is there any mechanism to adapt predictions based on actual execution results?

---

### Official Review · Reviewer_X2L8 · 2024-11-04

**Soundness:** 2
**Presentation:** 3
**Contribution:** 2
**Rating:** 3
**Confidence:** 2

**Summary:**

The paper proposes using LLMs to predict performance metrics of jobs submitted to a data center. Metrics include runtime, waiting time, and energy consumption. The main idea is to leverage the power of LLMs in creating meaningful representations of complex data, such as the source code, which can then be trained into specific predictions. Based on these predictions, the authors propose two scheduling algorithms, which they apply to a data center use case to show savings of up to approximately 30%.

**Strengths:**

+ The idea to leverage an LLM to create a powerful representation instead of hand-crafted features. This also brings a series of positive properties, as listed in the paper.
+ Good results on the presented data center use case.
+ Includes discussions on practical problems in applying the scheme.

**Weaknesses:**

- Lack of details on what is considered a job and its source code. Implicitly (especially in the introduction), authors seem to assume Python scripts for machine learning, but real-world workloads might differ from this assumption.  Please be more clear on any assumptions and restrictions on the jobs considered.
- Authors estimate the model performance metrics solely from the source code. Still, the execution time (and all other performance metrics) for some models, e.g., for LLMs due to their autoregressive nature, depends heavily on the generated output based on the input/prompt and not only on the source code. (see also next point)
- The paper lacks results on the achieved prediction accuracy of the considered metrics. Here, it would be nice to have some statistics on the achieved error between predicted and real values as well as a comparison with some of the mentioned related works for job prediction.
- Also, an ablation study to see if the improvement stems more from the smarter scheduling algorithm or from the more precise predictions would have been nice. This would also likely allow to hint into how well the approach might generalize to other data centers that might use a different baseline scheduling.

**Questions:**

What is the prediction error for runtime and energy consumption (also in comparison with baselines)?

It is well known that the lack of (remaining) run time of jobs is one of the main issues in achieving schedulings that are proven optimal under some aspects, i.e., minimizing waiting time. Can some variant of shortest job first (SJF) be used here?

Difference between simple and greedy alsgotiehms should be better highlighted.

---

### Note · Authors · 2024-11-28

I have read and agree with the venue's withdrawal policy on behalf of myself and my co-authors.